# A Bionic Sensing Platform for Cell Separation: Simulation of a Dielectrophoretic Microfluidic Device That Leverages Dielectric Fingerprints

**DOI:** 10.3390/biomimetics10110753

**Published:** 2025-11-07

**Authors:** Reza Hadjiaghaie Vafaie, Elnaz Poorreza, Sobhan Sheykhivand, Sebelan Danishvar

**Affiliations:** 1Department of Electrical Engineering, University of Bonab, Bonab 5551761167, Iran; rzvafaiee@gmail.com (R.H.V.); elnazpoorreza@gmail.com (E.P.); 2Department of Biomedical Engineering, University of Bonab, Bonab 5551761167, Iran; 3College of Engineering, Design, and Physical Sciences, Brunel University London, Uxbridge UB8 3PH, UK

**Keywords:** bionic, dielectrophoresis separation, biomedical, cancer, circulating tumor cells, microfluidics

## Abstract

Cancers are diseases described by the irregular spread of cells that have developed invasive features, enabling them to invade adjacent tissues. The specific diagnosis and effective management of oncological treatments depend on the timely detection of circulating tumor cells (CTCs) in a patient’s bloodstream. One of the most promising approaches to CTC separation from blood fractions involves the dielectrophoresis (DEP) technique. This research presents a new DEP-based bionic system designed for MDA-MB-231 breast cancer cell isolation from white blood cell (WBC) subtypes with a viable approach to cell viability. This work leverages the principle that every cell type possesses a unique dielectric fingerprint. This dielectrophoresis microfluidic device is designed to act as a scanner, reading these fingerprints to achieve a continuous, label-free separation of cancer cells from blood components with a high efficiency. In the proposed system that consists of three different stages, the first stage allows for separating B-lymphocytes and Monocytes from Granulocytes and MDA-MB-231 cells. The separation of B-lymphocytes from Monocytes occurs in the second step, while the last step concerns the separation of Granulocytes and MDA-MB-231 cells. In the analysis, x-y graphs of the electric potentials, velocity fields, pressure distributions, and cellular DEP forces applied to the cells, as well as the resulting particle paths, are provided. The model predicts that the system operates with a separation efficiency of nearly 92%. This work focuses on an investigation of the impact of electrode potentials, the velocity of cells, the number of electrodes, the width of the channel, and the output angles on enhancing the separation efficiency of particles.

## 1. Introduction

An expedited, highly sensitive, and minimally invasive technique for the isolation of circulating tumor cells (CTCs) from a peripheral blood draw is critically needed to propel advancements in biomedical oncology. These are cells scattered from the main tumor site into the patient’s bloodstream that are identified with cancer [1]. Measuring the number of cancer cells in the blood is highly important for both early cancer detection and for determining the effectiveness of surgery treatments. Consequently, the processes of isolating and concentrating CTCs from blood specimens has emerged as a critical focus of scholarly investigation [1].

Microfluidic technology has been developed as a revolutionary method proposing the capability of recapitulating macroscale physiological conditions on a small scale by the meticulous manipulation of fluids [2,3,4,5]. Microfluidic dielectrophoretic (DEP) devices, shaped as “Lab-on-a-Chip” (LoC) systems, could present a practical substitute for inspecting hematological cell structure in diagnostic applications. Equally, they could enable the gentle separation of cells for medical research, diagnostics, and therapeutic interventions [6,7,8]. DEP is a phenomenon described by the movement of electrically polarizable particles as a result of the dielectrophoretic force when subjected to a spatially varying electric field [9]. The polarity and intensity of the dielectrophoretic force exerted on cellular structures depend on the inherent electrical and geometric characteristics of the cells [10]. In light of the recent progressions in microfluidic technology, various microfluidic systems featuring different designs have been engineered to effectively sort, separate, and purify microparticles according to their dimensional disparities [11]. In general, the working mechanisms comprise the usage of microstructures [12], fluidic control [13], or external forces [14,15] to do these tasks [16].

While numerous methods of cell separation exist, dielectrophoresis has been found to be a highly effective approach, with its label-free nature and ability to perform differential discrimination between cell types based on their intrinsic properties. The principle behind DEP can be illustrated by an extremely intuitive example: in the same way that everyone possesses an exclusive fingerprint for identification, each cell type possesses an exclusive dielectric fingerprint. This fingerprint is not a static pattern, but is a dynamic electrical signature defined by the cell’s specific membrane capacitance and cytoplasmic conductivity. It is encoded within the cell’s frequency-dependent Clausius–Mossotti (CM) factor. In a DEP system, the application of a non-uniform electric field at a specific frequency acts as a scanner that reads this cellular fingerprint. Depending on the match between the applied frequency and the cell’s intrinsic dielectric profile, the cell will be either attracted towards the regions of high field intensity (positive DEP) or repelled from them (negative DEP). By carefully tuning the applied frequency and field geometry, these differential forces can be harnessed to physically guide distinct cell populations along separate paths within a microchannel. It is this principle of differential manipulation based on intrinsic properties that enables the precise, label-free sorting of a target cell (e.g., a CTC) from a background of other similar-sized cells (e.g., white blood cells) in a complex sample. This force is the response of the system to the detected identity. This is the principle that allows highly complex biological samples to be discriminated with a precise label-free accuracy.

The successful enrichment of CTCs and maintaining cell survival rate are imperative for subsequent analytical procedures. It is important to note that the electric field generated by microelectrodes, in combination with the dielectrophoretic forces, serves to localize the target cells within the specified areas [16,17,18,19,20]. The magnitude of the electric field tends to be significantly elevated in proximity to the electrodes. The passage of a cell into a high-field zone induces a brief loss of plasma membrane integrity. This compromise facilitates the efflux of cytoplasmic constituents, including organelles and nucleic acids [21]. The underlying principle is referred to as electroporation. The electroporation technique finds widespread application in numerous fields such as drug administration, cancer treatments, and nucleic acid-based therapy. However, cell lysis throughout the separation process could lead to an accidental dispersal of inner cell parts via an unexpected path. Therefore, in the area of dielectrophoresis-assisted separation, it is vital to reduce the electric field applied across the microchannel to achieve an efficient separation without prompting cellular disruption [21].

The Varmazyari group utilized FEM simulations to analyze the flow field and particle tracking alongside the resulting thermal loads from Joule heating, confirming that the semi-circular electrode configuration provided an optimal performance for the separation of CTCs from WBC subtypes [21]. The Uddin group proposed a label-free, high-output CTC isolation in a spiral microchannel via coupled dielectrophoresis and inertial forces [22]. The Sarowar group introduced a novel curved contraction–expansion microfluidic device equipped with DEP force, designed to achieve a high-purity separation of CTCs from the leukocyte background, overcoming the challenges related to size overlap and low CTC concentrations [23]. The Nguyen group utilized an effective microfluidic system to integrate a specialized DEP structure. This work leverages an FEC-DEP microstructure to achieve the separation of CTCs from various blood components, including erythrocytes, leukocytes, and platelets [24].

This research presents a bionic, DEP-driven microfluidic device engineered for the isolation of the MDA-MB-231 breast adenocarcinoma cell line from different white blood cell subtypes. This work introduces a novel paradigm for DEP-based separation: we treat the unique, frequency-dependent dielectric properties of a cell not as a single parameter, but as its distinctive “dielectric fingerprint.” The novelty of our approach lies in a microfluidic architecture specifically designed to interpret this rich fingerprint. Our three-stage device acts as a sophisticated scanner, where each successive stage is tuned to a different spectral feature of the cellular fingerprints, enabling the continuous and precise discrimination of cancerous cells from disparate white blood cell variants. In this device, which consists of three discrete phases, the preliminary phase eases the separation of B-lymphocytes and Monocytes from the cancerous MDA-MB-231 cell line and non-cancerous Granulocytes. The following phase includes the separation of B-lymphocytes from Monocytes, whereas the Granulocytes and MDA-MB-231 cell line are separated in the concluding phase. The analysis provides graphical depictions of the electric potentials, velocity fields, pressure distributions, and DEP forces applied to the cells, and the trajectories of the particles. The study focuses on evaluating the influence of electrode potentials, cell velocity, the number of electrodes, the width of the channel, and the output angles on the optimization of particle separation efficacy.

## 2. Methods

### 2.1. Theoretical Background of DEP

When polarizable particles suspended in a medium are subjected to a non-uniform spatial electric field, these particles experience a kind of directional movement driven by the action of dielectrophoretic forces. In the case of spherical particles, a precise and quantitative formulation that delineates the intensity of the DEP force was expressed through the mathematical representation given in Equation (1) [25,26,27,28,29,30,31,32,33]:(1)$$F_{DEP}=2\pi r^{3}\varepsilon_{0}\varepsilon_{m}Re[CM\left(f\right)]\nabla \left|E^{2}\right|$$

Here, *r* is considered to be the radius of a particle involved, and $\varepsilon_{m}$ and $\varepsilon_{0}$ are the variables invoked in this relation, representing the electrical permittivity of the surrounding environment and the permittivity of free space. E specifies the electric field intensity. The term that was introduced in this equation now gives the real component of the Clausius–Mossotti function, *Re*[*CM*(*f*)], and the second term signifies the square of the modulus of the electric field gradient, $\vec{(\nabla }|E^{2}|)$, that quantizes the spatial variation in the electric field strength around the particle. It must be noted that the Clausius–Mossotti factor takes values in the range from −1/2 to +1, and its sign specifically dictates whether positive or negative dielectrophoresis will occur in this system. Therefore, if CM > 0, the particle will exhibit a positive DEP and move in the direction of the maximum electric field strength, which is capable of usually being at the edge of an electrode. If, on the contrary, the Clausius–Mossotti factor is less than zero, CM < 0, the system will have a negative DEP force that exerts a repulsive force on the particles, driving them away from the highest intensity of the electric field.

The CM factor is defined as follows:(2)$$CM\left(f\right)=\frac{\varepsilon_{p}^{*}-\varepsilon_{medium}^{*}}{2\varepsilon_{medium}^{*}+\varepsilon_{p}^{*}},\;\varepsilon^{*}=\varepsilon -\frac{j\sigma }{\omega }$$

Here, $\varepsilon_{p}^{*}$ and $\varepsilon_{m}^{*}$ are the complex permittivity of the particle and the medium, respectively. They are, in turn, defined as follows [34]:(3)$$\varepsilon_{p}^{*}=\varepsilon_{p}-\frac{j\sigma_{p}}{\omega }$$(4)$$\varepsilon_{m}^{*}=\varepsilon_{m}-\frac{j\sigma_{m}}{\omega }$$
where $\sigma_{p}$ and $\sigma_{m}$ denote the particle and medium conductivity. Furthermore, the angular frequency is defined as ω = 2πf, *j* = $\sqrt{-1}$.

$Re\left[CM\left(f\right)\right]$ shows the real component of the CM factor, and the specific expression is defined as follows [34,35]:(5)$$Re\left[CM\left(f\right)\right]=\frac{\left(\varepsilon_{p}-\varepsilon_{m}\right)\left(\varepsilon_{p}+2\varepsilon_{m}\right)+\frac{\frac{\sigma_{p}-\sigma_{m}}{\omega }}{\frac{\sigma_{p}+2\sigma_{m}}{\omega }}}{({\varepsilon_{p}+2\varepsilon_{m})}^{2}+{\left(\frac{\sigma_{p}+2\sigma_{m}}{\omega }\right)}^{2}}$$
where *σ_p_* and *σ_m_* reveal the electrical conductivity of the particulate matter and the surrounding environment, respectively. Under conditions where ω is close to zero, special conditions arise [34,35].(6)$$Re\left[CM\left(f\right)\right]\approx \frac{\sigma_{p}-\sigma_{m}}{\sigma_{p}+2\sigma_{m}}$$

When ω→∞, the following is obtained:(7)$$Re\left[CM\left(f\right)\right]\approx \frac{\varepsilon_{p}-\varepsilon_{m}}{\varepsilon_{p}+2\varepsilon_{m}}$$

For simulating the cells, the single shell model is considered [36]:

The complex permittivity of the particle is denoted by $\varepsilon_{p}^{*}$, and that of the suspending medium by $\varepsilon_{med}^{*}$. For a biological cell, which is commonly modeled as a homogeneous core surrounded by a shelled membrane, the effective complex permittivity is given by the following expression [36,37]:(8)$$\varepsilon_{p}^{*}=\varepsilon_{mem}^{*}\frac{{\left(\frac{r}{r-d}\right)}^{3}+2(\frac{\varepsilon_{int}^{*}-\varepsilon_{mem}^{*}}{\varepsilon_{int}^{*}+2\varepsilon_{mem}^{*}})}{{\left(\frac{r}{r-d}\right)}^{3}-(\frac{\varepsilon_{int}^{*}-\varepsilon_{mem}^{*}}{\varepsilon_{int}^{*}+2\varepsilon_{mem}^{*}})}$$

In this expression, *d* represents the cell membrane thickness. The terms $\varepsilon_{int}^{*}$ and $\varepsilon_{mem}^{*}$ correspond to the complex permittivity of the interior cytoplasm and the surrounding membrane.

The single-shell model, depicting a spherical cell with its core and membrane, is shown schematically in Figure 1a. Figure 1b represents the conceptual model of the particle dynamics as a function of the DEP-induced movement in a spatially varying electric field for positive and negative DEP force.

### 2.2. Physics for Cell Separation

For the simulation of the proposed microseparator, COMSOL Multiphysics software (version 5.4) was utilized. This cell separation requires a detailed FEM simulation using three different modules: application of electric current, modeling of creeping flow, and tracing of particles in fluid flow. An approximation of fluid flow at very low Reynolds numbers, where the inertia term can be considered as negligible in the simulation, was obtained by using the module for creeping flow.

The fluid flow within the Lab-on-a-Chip microchannels was simulated under the creeping flow physics interface, which solves the Navier–Stokes equations under the negligible inertia term. This regime is known as Stokes flow and is described under a low Reynolds number. The governing equations are as follows [38]:(9)$$0=\nabla \cdot \left[-pI+K\right]+F$$
and(10)ρ∇·u=0

In these equations, *p* represents pressure, ***I*** shows the identity matrix, ***K*** is the viscous stress tensor, ***F*** is the volume force vector, ρ indicates the fluid density, and ***u*** denotes the velocity field [38].

The boundary conditions applied in this model are specified as follows: (a) The conditions at the microchannel walls within the apparatus conform to a non-slip boundary condition. (b) The boundary condition at both inlets was set to a prescribed normal inflow velocity. (c) A pressure boundary condition of zero was applied at the outlet.

Within this framework, the hydrodynamic resistance encountered by the cells is elucidated by Equation (11) [37]:(11)$$F_{drag}=-6\pi \mu rv$$

The parameters are defined as follows: dynamic viscosity (*μ*), cell radius (*r*), and cell velocity (***v***).

The motion of the cells is governed by Newton’s second law, which encompasses the governing equations of momentum and force. The Newtonian force balance equation for the cells is represented by Equation (12):(12)$$m_{cells}\left(\frac{\partial v}{\partial t}\right)=F_{drag}+F_{DEP}=F_{total}$$

We introduced the blood cells into the cell channel inlet based on their respective densities and diameters. A uniform distribution was selected at the inlets. The quantities of cells were initially maintained constant to enhance the visual representation of the particle trajectories.

In this examination, the electric current module is responsible for establishing and generating a non-uniform electric current field. An electric field was established within the fluid medium by applying dissimilar electrical potentials to the electrodes. The resulting charge density and electric field distribution were determined by solving Maxwell’s equations:(13)$$J=\sigma E+j\cdot \omega D+J_{e}$$

In this formulation, ***J*** denotes the current density, *σ* is the conductivity, ***E*** is the electric field, *j* is the imaginary unit, *ω* is the angular frequency, ***D*** is the electric displacement field, and ***J****_e_* is an external current source.

### 2.3. Geometry and Material Properties

Figure 2a presents a detailed schematic diagram of the microfluidic microseparator with rectangular electrodes precisely set in a line. The design of this separator is based on three different steps of operation. There are eight electrodes that are identical in both dimension and functionality in the first stage, while two and four electrodes are included within subsequent stages, since the second and third stages were added to the initial design in order to enhance the separation efficiency. The novel microseparator is proposed using two different inlets; one of those inlets plays the role of allowing the introduction of the blood sample, with dimensions precisely 200 µm in length and 40 µm in width [36], while the second inlet is used for the introduction of the buffer solution, also possessing similar dimensions to the blood sample inlet. This separator contains four separate outlets used for the separation of cells, each 200 µm in length and 40 µm in width, to ensure that the separation of the cellular components takes place as desired.

In order to provide the optimal focusing of the blood particles toward the electrodes, the inflow should be kept as high as 953 [µm/s] at the buffer inlet, which is notably higher than the inflow velocity at the blood sample inlet, recorded at 180 [µm/s] [36]. Table 1 presents a summary of the dielectric properties with respect to the four cell types, carrier fluid characteristics, and operational parameters of the study. First of all, selecting the appropriate frequency for the electrode voltage source is critical for the efficient separation of CTCs and other cell populations. Based on Table 1, the cell properties of all cell types are very close to each other. Generally, the separation of the cells is size-based. The reason for choosing the frequency of 70 [kHz] is that, in this frequency, the real parts of the Clausius–Mossotti factor for all cells are negative and the cells experience a negative DEP force; then, cell viability is guaranteed (Figure 2b). A negative DEP reduces the cell adhesion to electrodes, improving the long-term device stability. It must be noted that the positive DEP can cause CTCs to cluster near electrodes, complicating downstream analysis.

The cellular parameters used in this model, including cell radii, conductivities, and permittivities, were selected from the work of Varmazyari et al. [21] to maintain a consistent and validated baseline for simulation. It is noted that biological cell sizes can exhibit variability in the literature (based on the measurement technique, physiological state, and activation status of the cells used); however, this specific parameter set provides a well-defined and self-consistent framework for evaluating the separation performance of our proposed device architecture, which exploits the relative differences in these biophysical properties.

Our primary justification for selecting the MDA-MB-231 cell line is that it serves as a robust and clinically relevant model for aggressive, metastatic cancer cells. Successfully isolating such cells is of significant clinical importance. Compared to more epithelial-like breast cancer lines (e.g., MCF7, T47D), MDA-MB-231 cells are larger and have a more mesenchymal phenotype. This often results in different dielectric properties (specifically, lower membrane capacitance due to a less folded morphology), making them an excellent test case for demonstrating separation from WBCs based on these physical differences. The strength of our DEP-based approach is its tunability. This microseparator can be optimized for different CTC types by adjusting a key operational parameter such as the applied AC frequency based on the CM factor profile and other operating parameters such as geometry, applied voltage, and so on.

### 2.4. Mesh Set-Up

All the results were generated using a highly developed grid of computation, as depicted in Figure 3a, and the computation grid is made up of inner triangular elements that enable the modeling process. Care was taken to select a standard element size for the appropriate meshing of the proposed model. This mesh consists of 5972 domain elements presenting the volumetric accounts inside the model and 500 boundary elements constituting the interface of the domain from the surroundings. The grid independence study is conducted utilizing a varied number of mesh elements. Figure 3b,c illustrate the velocity profile in the x-direction and electric field intensity along the vertical cut line. As demonstrated in Figure 3b, the disparity in the channel’s velocity profile when employing a range of grid resolutions is insignificant. Therefore, a configuration of the channel comprising 5972 constituents with an average grid was chosen.

## 3. Results and Discussion

### 3.1. Validation and Assessment of the Model

To validate the computational model, its predictions were compared against the experimental data published by Piacentini et al. [40]. The comparison was specifically aimed at the trajectories of erythrocytes (RBCs) and platelets (PLTs), as indicated in Figure 4. Table 2 shows the dielectric properties of the PLTs and RBCs in the experimental work of Piacentini et al. [40].

The simulation was established with parameters that were commensurate with the experimental conditions, e.g., a voltage supplied by the electrodes of ±5 V, a cell injection velocity (v_in, cells) of 134 µm/s, and a buffer injection velocity of 853 µm/s (v_in, buffer). There was a very high correlation between the simulated path and the empirically recorded cell positions, confirming that the model is indeed valid, therefore confirming its prediction validity. It must be noted that Figure 4 and the related model validation section are intended solely to demonstrate the accuracy of our numerical solver and simulation methodology. In this section, we have precisely simulated the experimental conditions of the reference article of Piacentini et al. [40].

### 3.2. Simulated Physical Fields and Particle Dynamics: Spatial Electric Field, Flow Velocity, Pressure Path, and Force Due to Dielectrophoresis

Figure 5 represents a more detailed version of the different simulations concerning the electric potential distributions, velocity field, and pressure dynamics across the microseparator system. An extensive data analysis indicates that the regions of maximum electric potential, showing the maximum intensity, are highly localized adjacent to the electrode surface (Figure 5a). As shown in Figure 5a, the voltages applied to the first-stage, second-stage, and third-stage electrodes are 1.5, 2.5, and 1 V, respectively. Figure 5b shows the velocity field inside the microchannel. It should also be noted that the region characterized by the highest velocity field is located both at the inlet of the buffer velocity field and within the confines of the primary channel—a result that would suggest an important interaction between the dynamics at the inlet and the general behavior of the flow in the system. However, when dealing with the other stages of the microseparator, the intensity of the velocity field already possesses a weakening tendency. The buffer inlet velocity was set to 953 µm/s; in contrast, the cell inlet velocity was maintained at a significantly lower 180 µm/s. Note that the buffer is at a higher inlet velocity relative to the cells to enable the buffer to focus the cells closer to the electrodes for the highest interaction. A sheath flow for the buffer was introduced at a higher flow rate than the sample inlet to implement hydrodynamic focusing. This standard microfluidic technique uses laminar flow to confine the sample stream into a narrow path adjacent to the channel wall containing the electrodes. This ensures that all cells pass through the region of highest electric field gradient, thereby maximizing the dielectrophoretic force acting upon them and optimizing the separation efficiency.

Obviously, the maximum pressure point is situated not only at the entrance to the channel but also within the designed parameters of the principal channel itself, where the complex fluid dynamic inter-relations set in strongly due to a host of factors, as can be collected from Figure 5c. Consequently, it can be seen that a gradual and systematic loss of pressure is initiated, and this could only be attributed to the multi-interplay relationship between the flow characteristics and the resistance to flow across the enormous length of the channel. Lower internal pressures of the microchannel system are advantageous in maintaining blood cells’ structural integrity and shape, whereas higher pressures have been documented to apply significant deformations to these vital cellular structures [38].

Figure 6 and Figure 7 show the dielectrophoretic force experienced by each distinct cell species as they traverse the initial phase of the microchannel, an evaluation conducted over the duration of the simulation period, exactly analyzed along the vertical and horizontal red cut lines. It becomes obvious that B-lymphocytes and Monocytes show a relatively diminished response to the DEP force, as evidenced by the significantly lower values ranging from 11 × 10^−12^ [N] to 24 × 10^−12^ [N], as shown in Figure 6a,b, and 4.5 × 10^−12^ [N] to 9.5 × 10^−12^ [N], as depicted in Figure 7a,b. Conversely, in a similar analysis, it is observed that the MDA-MB-231 cells and Granulocytes exhibit the most pronounced susceptibility to the DEP force, which is verified by the notable elevation of the DEP force within the magnitude range of 75 × 10^−12^ [N] to 35 × 10^−12^ [N], as illustrated in Figure 6c,d, and 14 × 10^−12^ [N] to 30 × 10^−12^ [N], as reflected in Figure 7c,d, respectively. Additionally, it is noteworthy that the influence exerted by the DEP force persists for a duration of 4.5 s.

Figure 8 shows a detailed illustration of the separation of various cell types within the microseparator that has been suggested for this study. As clearly evidenced by the comprehensive simulation that is fully presented in Figure 7 and Figure 8, the MDA-MB-231 cells, which are depicted within dark red circles for clarity, as well as the Granulocyte cells, represented by green circles, experience the most substantial DEP force, a result of the electrodes being driven at 1.5 V, which consequently results in their deflection towards the lower outlets of the microseparator. Following this initial separation, the MDA-MB-231 cells and Granulocyte cells advance to the third stage, where a total of four active electrodes are engaged at a voltage of 1.2 V, which subsequently facilitates their exit through outlet 4 and outlet 3, respectively, thus ensuring an efficient separation process. Meanwhile, the B-lymphocytes, illustrated in dark blue, as well as the Monocytes, shown in blue, experience comparatively less deflection within the confines of the main channel, which leads them to transition to the second stage with a voltage of 2.5 V and ultimately exit via outlet 1 and outlet 2, respectively, highlighting the selective nature of the microseparator. Furthermore, all types of cells undergo negative DEP at a frequency of 70 kHz, a phenomenon that is critical to their movement. Due to the presence of a negative DEP, the cells are effectively repelled from the high-electric-field regions that are localized at the surfaces of the electrodes, thereby ensuring that the cells do not sustain any damage during the separation process. The design of a multi-stage separation system, with outlets for specific WBC subtypes, is a strategic choice to address the central challenge of dielectric property overlap in label-free CTC isolation. A single-stage device attempting to separate CTCs from a bulk WBC population would suffer from a significant contamination, as the dielectric signatures of Monocytes and B-lymphocytes can closely resemble those of various CTCs. Our sequential approach mitigates this by first removing the most dielectrically distinct populations (Stage 1), followed by a finer separation to resolve the overlapping subtypes (Stages 2 and 3). This ensures that the final CTC outlet is of high purity. Furthermore, this architecture provides the added potential to correlate CTC numbers with specific immune cell counts from the same blood sample, offering a more comprehensive liquid biopsy platform.

The damaging impacts on living cells by electric currents include Joule heating and electroporation. Joule heating results from the conversion of electrical energy into heat that results in localized heating and subsequent damage to the cells. Electroporation results in the formation of holes in the cell membrane by enhanced electric field strengths capable of damaging the structural integrity and viability of the cells. These effects are avoided by the optimization of the voltage applied in the DEP system for enhanced efficacy. For example, low-conductivity media (conductivity of the carrier sample fluid inside the microchannel) was used to minimize Joule heating, and the voltages were selected to reduce the risk of electroporation. Studies have shown that electric field intensities below 1–2 kV/m [21] are generally safe for short-term exposure (seconds to minutes). In the proposed microseparator, the electric field intensities were maintained within the range of 1 kV/m, which is well below the reported thresholds for cell damage [21].

Figure 9 illustrates the comprehensive computation of the electric field norm in the vicinity of the electrodes, as observed through the first stage presented in Figure 9a, the second stage in Figure 9b, and the third stage illustrated in Figure 9c, which has been evaluated over the simulation duration along the horizontal cut lines that have been marked with red lines for emphasis. It is noteworthy that the peak electric field intensity observed in close proximity to the electrodes during the second stage of the analysis surpasses that of the other stages, reaching an impressive value of approximately 1 × 10^5^ [V/m], a phenomenon that can be attributed to the elevated input voltage that has been specifically configured for the operation of the electrodes in question. One of the fundamental requirements in DEP separation is regulating the intensity of the applied electric field. Reducing the strength of the field along the microchannel is required to obtain effective separation without causing cellular damage such as lysis or electroporation. Based on our simulations, the electric field intensity used is safe for MDA-MB-231 and certain WBC subtypes.

During the simulation, capable of characterizing nearly 100 distinct WBC variants each second (which is equivalent to 3.6 × 10^5^ cells per hour) [21], 5 CTCs per second are entered into the cell inlet. This throughput corresponds to the cell count found in approximately 72 µL of a human blood specimen, which is predominantly composed of white blood cells with a minimal population of circulating tumor cells [21]. Within the system, cells are transported through the microchannel in 4.5 s and are subsequently collected at the designated outlets. The efficiency of separation is computed based on the simulated concentration of CTCs, defined as follows:

Separation Efficiency = (Number of CTCs in intended output/Total count of CTCs introduced) × 100%. The efficiency of the proposed microseparator can be calculated close to 92%.

The use of singular mean values represents an idealized scenario, and accounting for the natural size distribution of cells is essential for predicting realistic device performance.

It must be noted that the initial approach of using singular mean values is a common and well-established procedure in computational studies to provide a first-order comparison of device designs and to demonstrate the fundamental feasibility of a separation concept. The 92% efficiency reported under these conditions serves as a benchmark for the theoretical maximum performance of our device architecture.

A comparative study of the proposed DEP-based device with other technologies is summarized in Table 3. Note that the studies referred to here utilized different operational parameters for CTC separation. The maintenance of cell viability in the DEP systems is a critical factor that involves handling Joule heating appropriately. This occurrence is due to the usually high local electric fields close to the microelectrodes. To avoid this consequence, our design uses two significant parameters: applying reduced operating voltages and utilizing a buffer with a minimized electrical conductivity (kept below 100 mS/m). Therefore, this work exhibits an effective cell separation with the additional consequence of reduced Joule heating. In conclusion, the novelty of our microseparator as a dielectric fingerprint scanner is its ability to achieve a high-efficiency separation at a low operating voltage (<2.5 V), combined with a multi-stage architecture that ensures a high purity by resolving dielectrically overlapping cell populations. Many DEP devices in the literature require higher voltages (often 5–10 Vpp) to generate sufficient field gradients for effective separation. This combination of gentle operation and high performance represents a significant advancement over the existing DEP devices.

Figure 10 shows the temperature profile across the channel length calculated separately for three microchannels. It is clear that the temperature gradient along the microchannel is approximately 1 K, which maintains the cell solution within a safe range to ensure viability [21].

### 3.3. The Effect of Changing the Input Voltage on the Trajectories of the Cells

A key relationship in dielectrophoresis is that the force is proportional to (∇E^2^), which is directly influenced by the applied voltage. At higher voltages, particles experience stronger DEP forces, making it easier to separate them based on their dielectric properties. However, excessively high voltages can cause unwanted effects like Joule heating or particle damage. Based on our simulations, a higher applied voltage produces a stronger DEP force on the cells.

Figure 11a,b show the simulation of cell trajectories when a variable voltage of (a) 1 V and (b) 2 V is used for the first stage. When an input voltage of 1 V is considered, this voltage derives a relatively weak DEP force that deflects all the cells to the second stage. Applying 2 V results in dynamics ensuring that all the cells go to the third stage. It can be seen that, in both cases of voltage change, successful separation does not occur. Figure 11c,d show the simulation of cell trajectories when a variable voltage of (a) 2 V and (b) 3 V is used for the second-stage electrodes. In these cases, both the B-lymphocytes and Monocytes exit from a similar outlet. Figure 11e,f show the simulation of cell trajectories when a variable voltage of (e) 1 V and (b) 1.5 V is utilized for the third-stage electrodes. In these cases, both Granulocytes and MDA-MB-231 cells exit from an identical outlet. Therefore, in these circumstances, the separation of the cells fails to occur.

### 3.4. The Effect of Changing Velocity of Cells on the Cell Trajectories at the Outlet

Higher flow velocities can reduce the time cells spend in the DEP field, potentially decreasing the effectiveness of DEP forces in separating cells based on their dielectric properties. Cells may not have enough time to migrate to their equilibrium positions. Lower flow velocities allow cells more time to experience DEP forces, improving the separation efficiency. However, excessively low velocities may lead to clogging or nonspecific interactions. Excessively low velocities reduce the shear forces that keep cells in suspension, increasing the risk of nonspecific cell-surface adhesion and cell–cell aggregation. These events can lead to channel clogging and fundamentally alter the dielectrophoretic behavior of the cells, thereby degrading the separation performance. An optimal velocity ensures that cells are exposed to the DEP field for a sufficient duration, allowing for a clear differentiation between cell types. High flow velocities can subject cells to shear stress, potentially damaging them or affecting their viability. This is particularly important for sensitive cell types. Lower velocities are generally gentler on cells, but they may increase the risk of cell sedimentation or aggregation. The velocity field interacts through the application of DEP forces to the cells. If the hydrodynamic drag force (due to flow velocity) dominates the DEP force, cells may not respond effectively to the DEP field. High velocities can potentially reduce nonspecific interactions (e.g., sticking to walls of channels) but also potentially create turbulence or mixing that will disrupt separation. Low speeds have the potential to increase the likelihood of nonspecific interactions, i.e., sticking or clumping.

Figure 12 shows a simulation of the changing velocity of cells on cell trajectories when the velocity is fluctuated between (a) 80 [μm/s], (b) 280 [μm/s], and (c) 380 [μm/s]. As indicated in Figure 11a, when the cell velocity is set at 80 [μm/s] during a time interval of 3.5 s in the third stage, the MDA-MB-231 cells and Granulocytes exit from similar outlets. When the velocity changes to 280 [μm/s] (Figure 12b) in the second stage, B-lymphocytes exit from outlets 1 and 2. In the case of Figure 11c, when the cell velocity is increased to 380 [μm/s], due to the increased velocity, B-lymphocytes and Monocytes exit from the identical outlet in the second stage (outlet 2), while Granulocytes go to outlet 1.

### 3.5. The Effect of Changing the Number of Electrodes on the Trajectories of the Cells and DEP Force

In this part, the impact of modifying the number of electrodes is investigated. The number of electrodes in a DEP-based microseparator significantly impacts the trajectories of cells and the distribution of DEP forces. More electrodes increase the spatial resolution of electric field gradients, enabling finer control over DEP forces. This allows for more complex field patterns, enhances the uniformity or specificity of force distribution, and improves separation efficiency. Having fewer electrodes simplifies the electric field but may lead to weaker or less precise DEP forces. Increasing the number of electrodes generally enhances DEP force precision and cell trajectory control, improving the separation resolution. However, optimization is needed to balance performance with fabrication and operational complexity. Figure 13a shows the cell trajectories when the number of first-stage electrodes is reduced from eight to four. By a reduction in the number of electrodes, the sufficient DEP force does not exist to push the cells toward the third stage, and all the cells go to the second stage and exit from outlet 1. In Figure 13b, the electrodes of the second stage have been deleted. In this case, in the absence of the electrodes and the corresponding DEP force, both the B-lymphocytes and Monocytes exit from outlet 2. Figure 13c shows the simulation of cell trajectory when the electrodes of the third stage have been reduced from four to two. In this condition, with the reduction in the DEP force, both Granulocytes and CTCs exit via outlet 3. As can be seen, by changing the number of electrodes, the desired goal of the isolation of cells cannot be achieved in all cases.

### 3.6. The Effect of Changing the Width of First, Second, and Third Stages on the Cell’s Trajectories and DEP Force

The width of the microfluidic channel in a DEP separation system can significantly influence cell trajectories due to its impact on the electric field distribution and dielectrophoretic forces. In thinner channels, the electric field gradient is typically stronger and more uniform across the channel height. This can lead to more predictable and consistent DEP forces acting on the cells, resulting in well-defined trajectories. In wider channels, the electric field gradient may weaken or become less uniform, especially farther from the electrodes.

Figure 14 shows the cell trajectories when the thickness for the first, second, and third stages increases to 50 μ (original thickness was 40μ). In this case, by increasing the thickness and reducing the DEP force, all cells move to the second stage, and the preferred separation is not achieved.

### 3.7. The Effect of Changing the Input Angle (α) and Output Angles (β, γ) on the Cell Trajectories

In this section, the influence of the variability of both the input and output angles on cell trajectories is investigated. Figure 15 shows the effect of changing the input angle (α) in the first and second (or third) stages when (a) α>90° and (b) α<90°. As is clear from Figure 15a, the angle change leads to the entrance of the Granulocytes to the second stage, and this results in an unsuccessful separation of the cells. In Figure 15b, the adjustment of the angle does not yield any appreciable alterations in the paths taken by the cells. Figure 16 shows the simulation of the effect of changing the output angle (β,γ) when (a) β>80°, (b) β<80°, (c) γ>30°, and (d) γ<30°. The initial output angle of second and third stages is set to 80° and 30°, respectively. As is clear from Figure 16a, increasing the output angle to β>80° does not have any effect on the cell trajectory, but, in case when the angle changes to β<80° (Figure 15b), both Monocytes and B-lymphocytes simultaneously exit from the outlet (outlet 1). In Figure 16c,d, when the output angle of the third stage (γ) changes to (c) γ>30° and (d) γ<30°, the variation does not affect the trajectory of cells. It must be noted that, in Figure 16c, the Granulocytes do not reach the outlet in 3.5 s.

Table 4 presents the parametric analysis results, showing that electrode number and applied voltage have the greatest impact on separation efficiency. Increasing electrodes and voltage notably improves performance, achieving over 90% efficiency under optimized conditions. Other parameters such as flow velocity, channel thickness, and angles (α, β, γ) show minor influence, maintaining efficiencies around 90%.

Recent advances in artificial intelligence (AI), biomedical signal analysis, and smart sensing technologies have profoundly influenced the development of bionic and microfluidic systems. Deep learning and generative architectures have demonstrated an exceptional potential for improving diagnostic precision and feature extraction in biomedical imaging and biosignal applications [44,45]. The integration of machine learning with intelligent decision-making frameworks has also accelerated automation across healthcare diagnostics, treatment, and patient management [46,47]. Moreover, hybrid frameworks combining blockchains, AI, and fuzzy logic have introduced secure and explainable infrastructures for healthcare logistics and biomedical data handling [48,49,50]. Parallel developments in AI-based structural optimization and intelligent material design have extended computational intelligence from software to physical domains, enabling high-fidelity modeling and precision manufacturing in biomedical and bionic engineering [51,52]. These methods strengthen the predictive capability of microfluidic simulations, particularly for estimating the dielectric and electrokinetic characteristics of biological cells. Additionally, interdisciplinary research in cognitive modeling and human–machine interfaces [53,54] underscores the convergence of neuroscience, materials science, and bioengineering. Collectively, these studies provide a conceptual and technological foundation for the proposed bionic sensing platform—a dielectrophoretic microfluidic device capable of adaptive control and dielectric fingerprint-based cell separation [55].

## 4. Conclusions

The separation of particles, according to their intrinsic properties, is indeed the focus of recent investigations in biotechnology, medicine, and many other disciplines. Separation techniques for very small particles in the micrometer or nanometer range are at the very edge of current investigation, since many of the methods developed so far either suffer from serious flow limitations or a lack in selectivity. Dielectrophoresis represents a methodology that can target various particle properties and thus can be put forward as a candidate technique for solving challenging separation tasks. In the present work, a microfluidic chip, designed by using the finite element method, was simulated for its performance in blood cell sorting. The goals of the present study are the separation of MDA-MB-231 cells along subpopulations of white blood cells using the methodology of dielectrophoretic force in a microchannel. Our results show that the proposed microfluidic set-up is able to isolate cancer cells with the use of rectangular electrodes operating at a frequency of 70 kHz.

We have successfully modeled and validated a three-stage DEP microfluidic device that operates on the powerful principle of the “dielectric fingerprint.” By tuning electric fields to selectively read the unique biophysical signatures of different cell types, our system achieves a continuous, label-free isolation of MDA-MB-231 cells from white blood cells with an efficiency of 92%. This work confirms that leveraging these intrinsic cellular fingerprints is a robust and effective strategy for complex bio-separation tasks. A comprehensive statistical simulation accounting for full polydisperse populations is an essential and planned next step for experimental validation and further device optimization in future work.

Looking forward, the fingerprint paradigm opens a clear path for next-generation microfluidic systems. Future work will focus on developing “adaptive” or “cognitive” separators that can dynamically adjust frequencies in real time to identify and isolate rare cells based on their unique dielectric profiles, moving us closer to truly intelligent, bionic diagnostic platforms.

At the end of this study, we will be in a position to provide useful insights for researchers to take into consideration while designing systems before performing biological experiments.

## Figures and Tables

**Figure 1 biomimetics-10-00753-f001:**
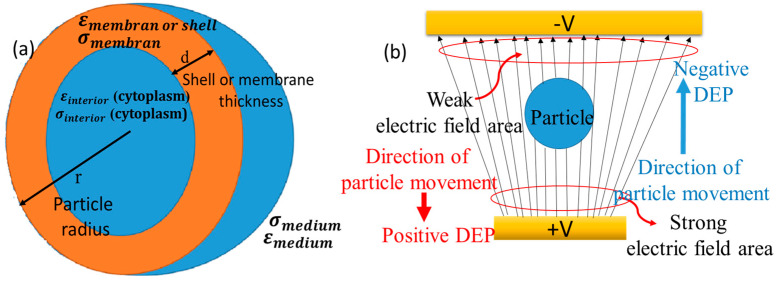
(**a**) Geometric representation of a cellular structure using the one-shell model. Schematic of dynamics of particle (**b**) movement under the influence of dielectrophoretic responses in the context of a spatially varying electric field.

**Figure 2 biomimetics-10-00753-f002:**
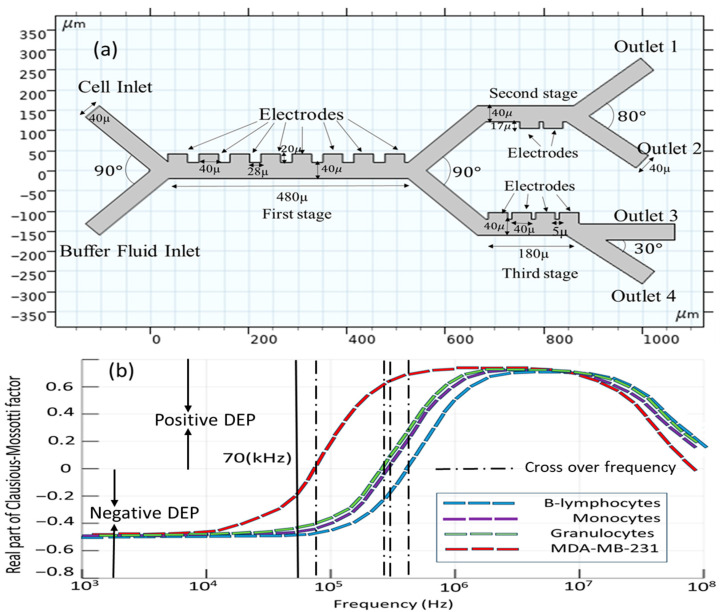
(**a**) Conceptual layout of the microfluidic chip; (**b**) real part of Clausios–Mossotti factor for the cells.

**Figure 3 biomimetics-10-00753-f003:**
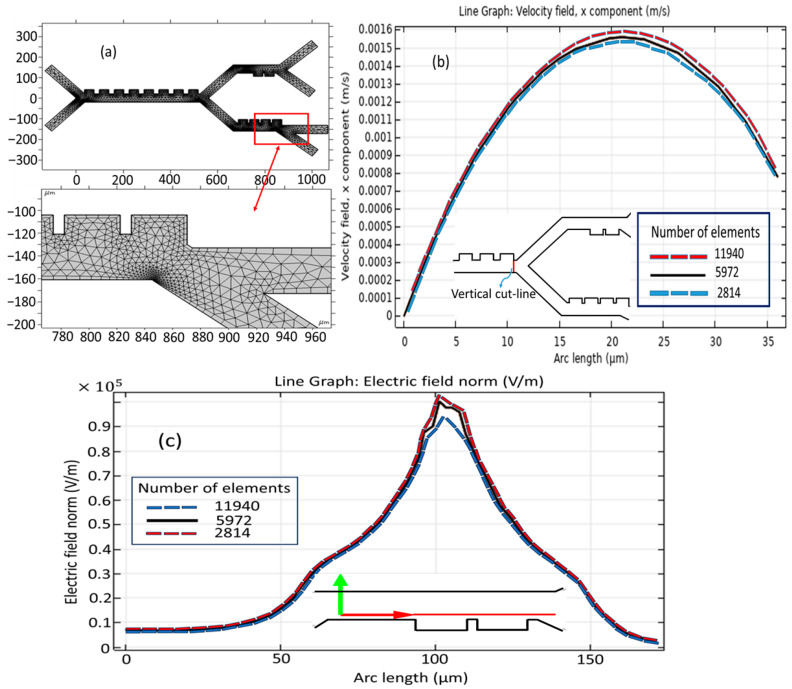
Mesh set-up: (**a**) meshing the microseparator; (**b**) grid independency study for velocity field and (**c**) for electric field across the second stage.

**Figure 4 biomimetics-10-00753-f004:**
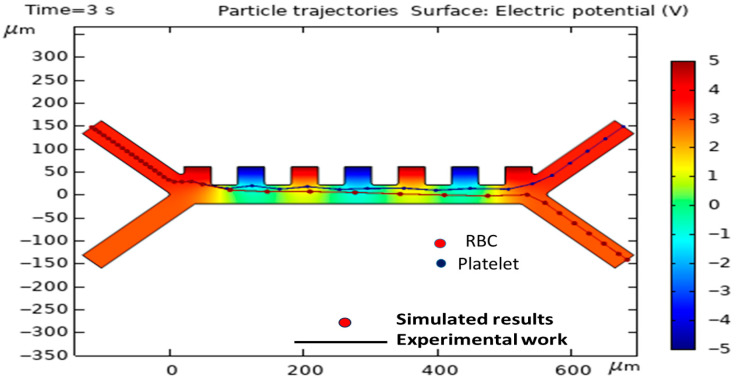
Agreement between the simulated trajectories of RBCs and platelets (PLTs) and the empirical data from Piacentini et al. [40].

**Figure 5 biomimetics-10-00753-f005:**
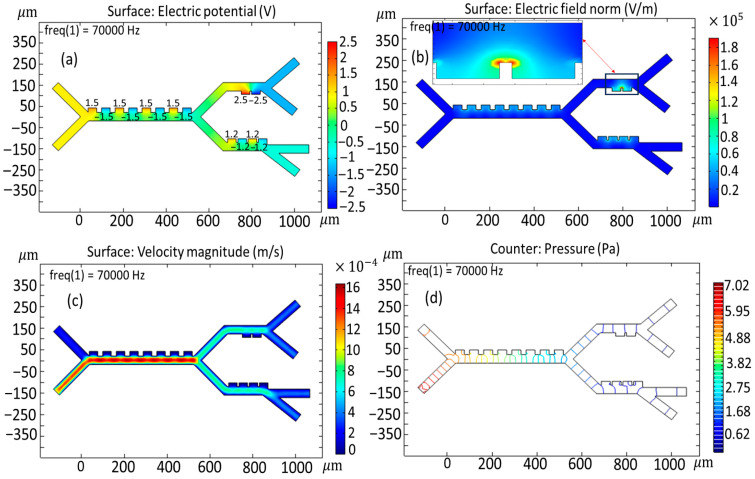
Key simulation outputs for the proposed microseparator: (**a**) potential field, (**b**) intensity of electric field, (**c**) velocity field, and (**d**) pressure distribution.

**Figure 6 biomimetics-10-00753-f006:**
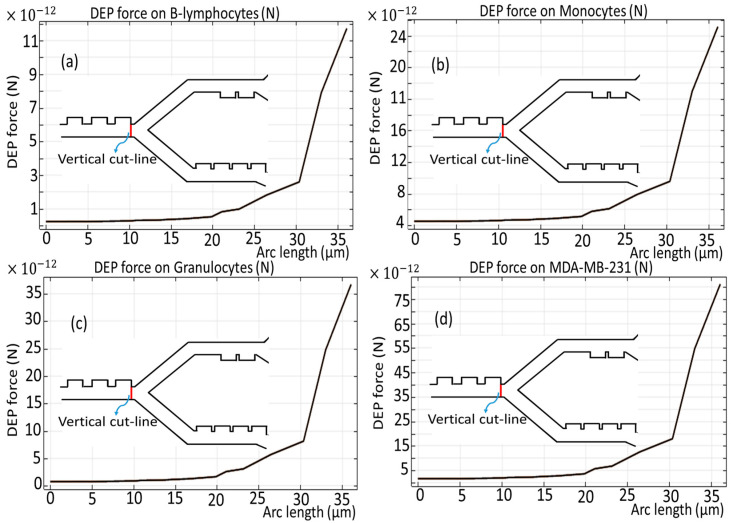
DEP force for the cells through the first stage across the vertical cut line presented as a red line.

**Figure 7 biomimetics-10-00753-f007:**
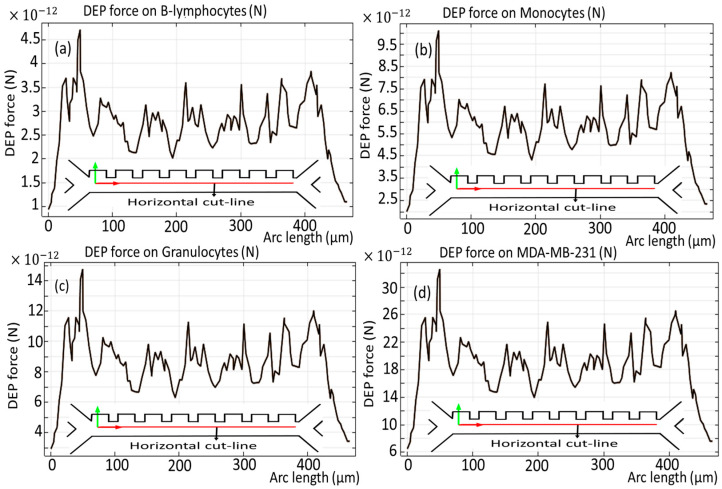
DEP force for the cells through the first stage of the microchannel across the horizontal cut line.

**Figure 8 biomimetics-10-00753-f008:**
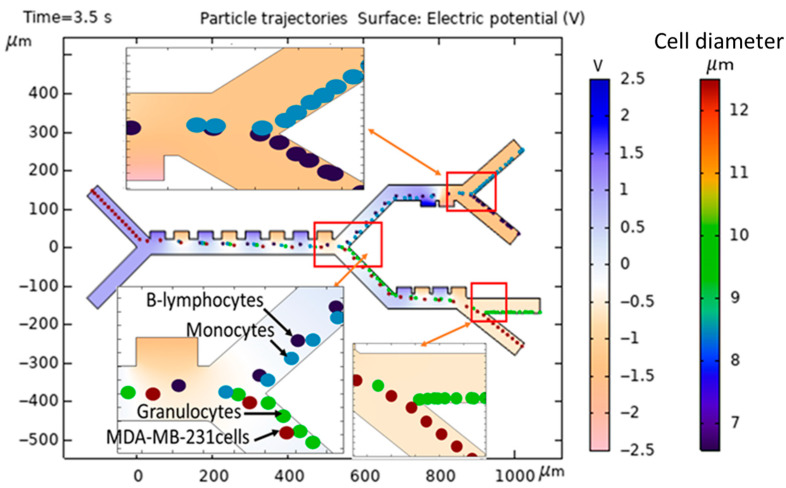
Modeling and simulation of separation of MDA-MB-231 cancer cells from the subtypes of WBCs in the proposed microseparator.

**Figure 9 biomimetics-10-00753-f009:**
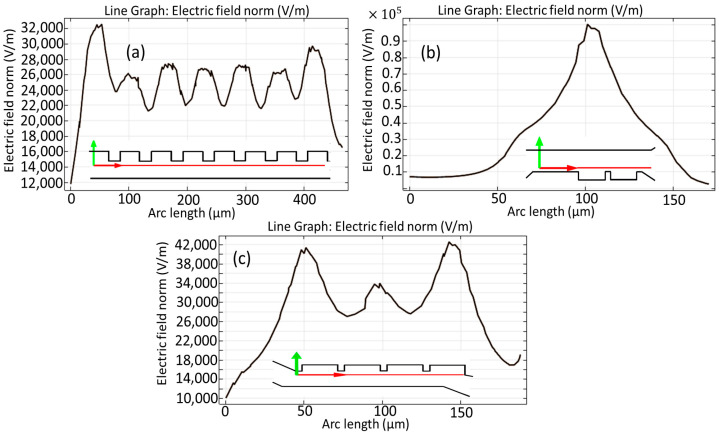
Calculations of the electric field norm around the electrodes during the first step (**a**), second step (**b**), and final step (**c**) of the microchannel over the horizontal cut lines in red.

**Figure 10 biomimetics-10-00753-f010:**
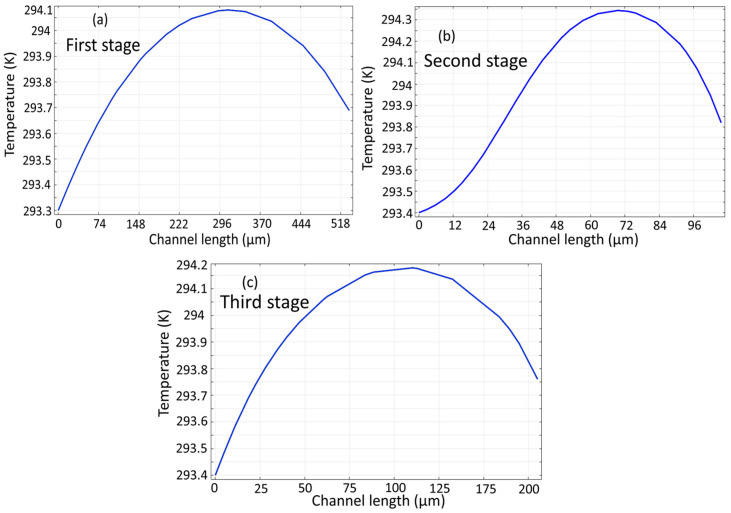
Temperature profile of the proposed microseparator: (**a**) initial stage, (**b**) subsequent second stage, (**c**) and final stage.

**Figure 11 biomimetics-10-00753-f011:**
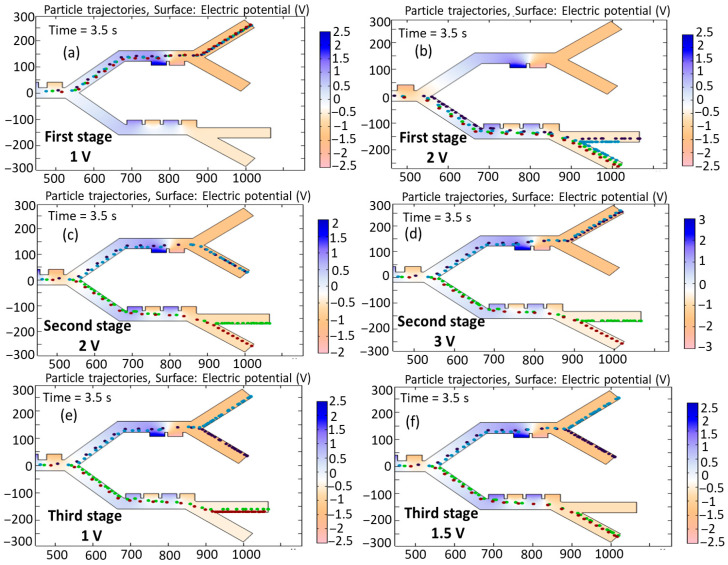
Simulation of the influence of changing the input voltage on the trajectories of the cells: first stage: (**a**) 1 V, (**b**) 2 V; second stage: (**c**) 2 V, (**d**) 3 V; and third stage: (**e**) 1 V, (**f**) 1.5 V.

**Figure 12 biomimetics-10-00753-f012:**
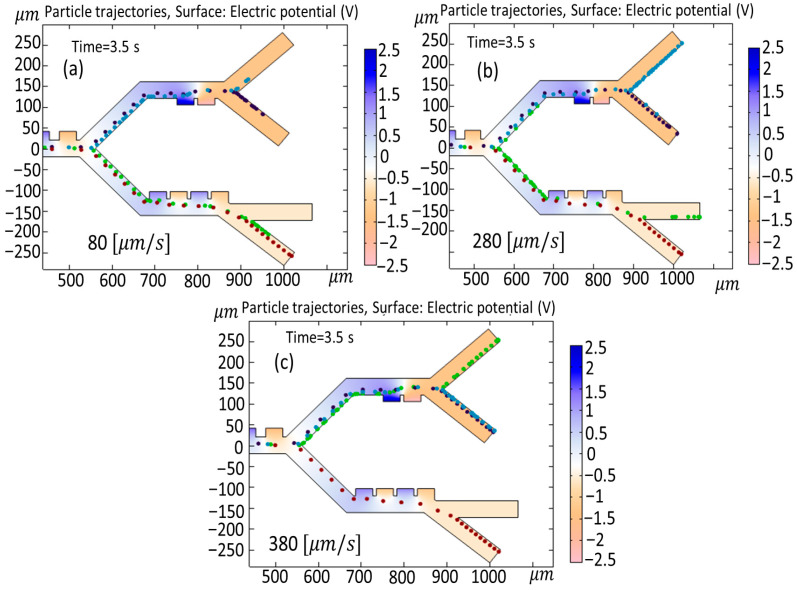
Simulation of the changing velocity of cells on cell trajectories when the velocity is fluctuated between (**a**) 80 [μm/s], (**b**) 280 [μm/s], (**c**) and 380 [μm/s].

**Figure 13 biomimetics-10-00753-f013:**
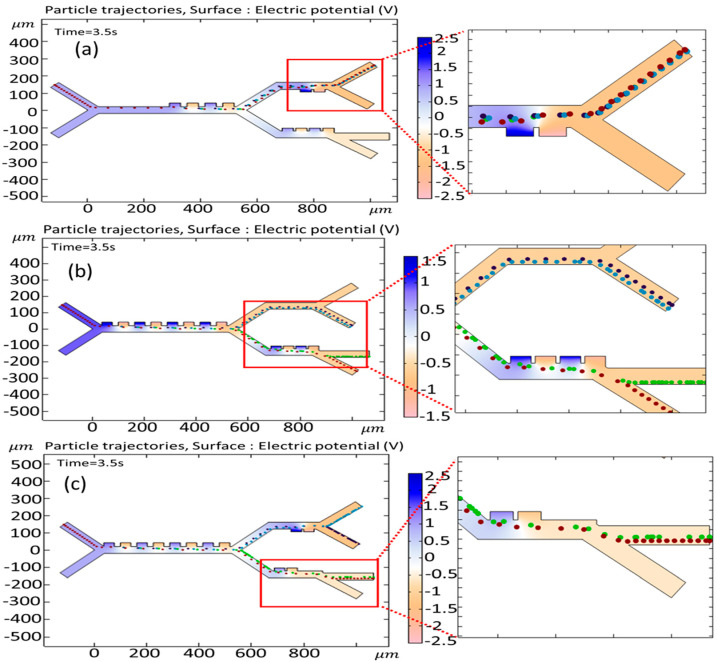
Simulation of the effect of changing the number of electrodes on the cell trajectories (**a**) when the number of first-stage electrodes is reduced from eight to four, (**b**) when the electrodes of the second stage have been removed, and (**c**) when the electrodes of the third stage have been reduced from four to two.

**Figure 14 biomimetics-10-00753-f014:**
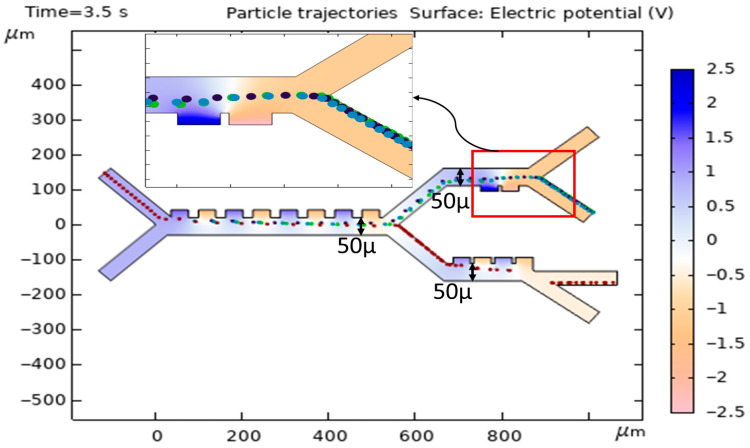
Simulation of the effect of increasing the thickness of the channel on the cell trajectories.

**Figure 15 biomimetics-10-00753-f015:**
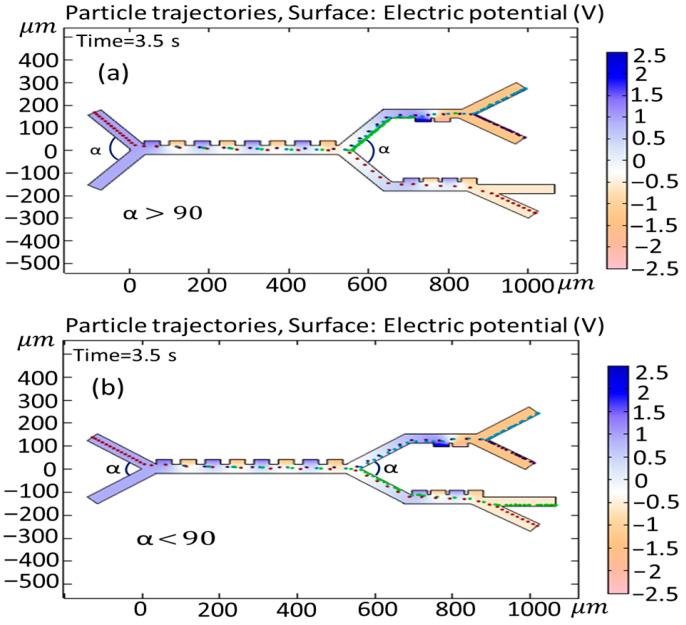
Simulation of the effect of changing the input angle (α) of the first and second stages, when (**a**) α>90° and (**b**) α<90°.

**Figure 16 biomimetics-10-00753-f016:**
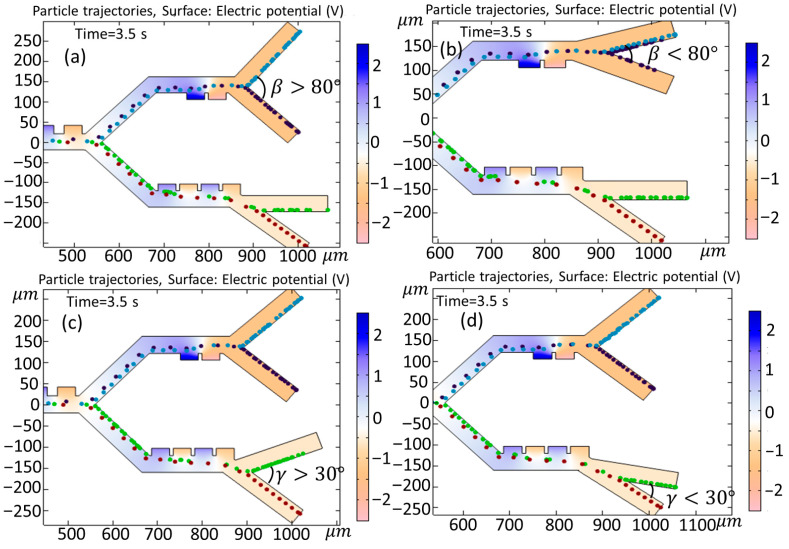
Simulation of the effect of changing the output angle (β.γ) when (**a**) β>80°, (**b**) β<80°, (**c**) γ>30°, and (**d**) γ<30°.

**Table 1 biomimetics-10-00753-t001:** The dielectric properties of the constituents and the operational parameters [21,39].

Parameter	Value
Frequency	70 [kHz]
Electrical conductivity of the suspension medium	55 [mS/m]
Electrical permittivity of the suspension medium	80
Fluid density	1000 [kg/m^3^]
Viscosity of the fluid	$1\times 10^{-3}$ [Pa*s]
Particle density	1050 [kg/m^3^]
Cell radius: Granulocytes	4.97 [µm]
Cell radius: B-lymphocytes	3.28 [µm]
Cell radius: Monocytes	4.23 [µm]
Cell radius: MDA-MB-231	6.2 [µm]
Cell conductivity: Granulocytes	0.6 [S/m]
Cell conductivity: B-lymphocytes	0.73 [S/m]
Cell conductivity: Monocytes	0.56 [S/m]
Cell conductivity: MDA-MB-231	0.6 [S/m]
Cell dielectric constant: Granulocytes	151
Cell dielectric constant: B-lymphocytes	154
Cell dielectric constant: Monocytes	127
Cell dielectric constant: MDA-MB-231	52
Conductivity of shell: Granulocytes	$1\times 10^{-6}$ [S/m]
Conductivity of shell: B-lymphocytes	$1\times 10^{-6}$ [S/m]
Conductivity of shell: Monocytes	$1\times 10^{-6}$ [S/m]
Conductivity of shell: MDA-MB-231	$1\times 10^{-6}$ [S/m]
Shell dielectric constant: Granulocytes	5
Shell dielectric constant: B-lymphocytes	5
Shell dielectric constant: Monocytes	5
Shell dielectric constant: MDA-MB-231	5
Shell thickness: Granulocytes	4 [nm]
Shell thickness: B-lymphocytes	4 [nm]
Shell thickness: Monocytes	4 [nm]
Shell thickness: MDA-MB-231	4 [nm]

**Table 2 biomimetics-10-00753-t002:** The dielectric properties of the PLTs and RBCs in the experimental work of Piacentini et al. [40].

Cell	Diameter	Conductivity	Relative Permittivity	Shell Conductivity	Shell Permittivity	Shell Thickness
PLT	1.8 [µm]	0.25 [S/m]	50	$1\times 10^{-6}$ [S/m]	6	8 [nm]
RBC	5 [µm]	0.31 [S/m]	59	$1\times 10^{-6}$ [S/m]	4.44	9 [nm]

**Table 3 biomimetics-10-00753-t003:** Benchmarking of the proposed DEP device against existing technologies.

Selected Cell Type	Buffer Conductivity (mS/m)	Frequency	Voltage (V)	Efficiency	References
MDA-MB-435, MDA-MB-468, MDA-MB-231	30	15–60 kHz	10 (Vpp)	90	[41]
MDA-MB-231, HeLa	1	100 kHz	7 (Vpp)	98	[42]
MDA-MB-231	55	1 kHz	10 (Vpp)	90	[43]
MDA-MB-231	55	125 kHz	3.52 (Vp)	95	[21]
MDA-MB-231	55	70 kHz	2.5 (Vp)	92	This study

**Table 4 biomimetics-10-00753-t004:** Summary of parametric analysis on separation efficiency.

Parameter	Condition/Value	Separation Efficiency
**Number of Electrodes (Stage 1)**	4 electrodes	2%
	8 electrodes	92%
**Number of Electrodes (Stage 2)**	0 electrodes	90%
	2 electrodes	91%
**Number of Electrodes (Stage 3)**	2 electrodes	5%
	4 electrodes	92%
**Applied Voltage (Stage 1)**	1.0 V	2%
	2.0 V	62%
**Applied Voltage (Stage 2)**	2.0 V	91%
	3.0 V	92%
**Applied Voltage (Stage 3)**	1.0 V	11%
	1.5 V	82%
**Flow Velocity**	80 µm/s	81%
	280 µm/s	90%
	380 µm/s	91%
**Channel Thickness**	40 µm	92%
	50 µm	5%
**Input Angle (α)**	α > 90°	91%
	α < 90°	90%
**Output Angle (β)**	β > 80°	92%
	β < 80°	92%
**Output Angle (γ)**	γ > 30°	90%
	γ < 30°	92%

## Data Availability

The raw data supporting the conclusions of this article will be made available by the authors on request.

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
