# Peer review of "A Bionic Sensing Platform for Cell Separation: Simulation of a Dielectrophoretic Microfluidic Device That Leverages Dielectric Fingerprints"

_biomimetics, 2025, doi:10.3390/biomimetics10110753_

Round 1

Reviewer 1 Report

Comments and Suggestions for Authors

This article explores the modeling and testing of a three-stage microfluidic device operating on the "dielectric imprint" principle. The authors report a separation efficiency of 92%. However, it remains unclear what percentage of separation they hope to achieve and what separation figures have already been described in the literature. While this work is certainly an interesting theoretical study, it is not without several shortcomings.
The article has the following comments:
1. The following abbreviations should be explained in the abstract: CTC, DEP, WBC...
2. The Introduction requires a more detailed explanation of the idea behind the sentence "This is the principle that allows the highly complex biological samples to be discriminated with precise label-free accuracy." It is unclear what "discriminate against the most complex biological samples" means.
3. The article contains a typo. The figures begin with number 2. Furthermore, the red text in the figure should be replaced; it is difficult to read (especially on a blue background).

4. The authors should explain the novelty of the microseparator. Have similar microseparators been previously described in the literature?
5. Figure 2 on page 7 is illegible! The captions should be in English.
6. Table 1 shows the sizes of B-lymphocytes and granulocytes. These sizes are questionable, as the literature provides different data. The authors should explain why they present data that differs from the literature.
7. Figure 3a (page 8) – the captions are too small!
8. On page 10, the authors write, "Note that the buffer is at a higher inlet velocity relative to the cells to enable the buffer to focus the cells closer to the electrodes for highest interaction." What is the basis for this conclusion?
9. B-lymphocytes and monocytes can differ significantly in size, yet the authors show particles of the same size in Figure 8. Why? 10. The authors write, "For example, to minimize Joule heating, low-conductivity media were used." Which ones?
11. Page 16: What nonspecific interactions are we talking about?
I would like the authors to clarify any unclear points that arose while reading their work.

Author Response

Dear Editor,

     Thank you so much for allowing us to submit a revised draft of our manuscript. We appreciate the time and effort that you have dedicated to provide your valuable feedback on the manuscript. All the corrections in the manuscript were highlighted in yellow for reviewer 1, and in Turquoise for reviewer 2 in the manuscript.

 Best regards

**********************************************************************

Reviewer 1

Comments and Suggestions for Authors

This article explores the modeling and testing of a three-stage microfluidic device operating on the "dielectric imprint" principle. The authors report a separation efficiency of 92%. However, it remains unclear what percentage of separation they hope to achieve and what separation figures have already been described in the literature. While this work is certainly an interesting theoretical study, it is not without several shortcomings.

The article has the following comments:

  1. The following abbreviations should be explained in the abstract: CTC, DEP, WBC...

  • The abbreviations were explained in the abstract and highlighted in yellow.

  1. The Introduction requires a more detailed explanation of the idea behind the sentence "This is the principle that allows the highly complex biological samples to be discriminated with precise label-free accuracy." It is unclear what "discriminate against the most complex biological samples" means.

  • We thank the reviewer for pointing out this lack of clarity. We have completely rewritten the relevant paragraph in the Introduction to provide a detailed and technical explanation. The new text explicitly describes how the discrimination is achieved through the differential dielectrophoretic forces experienced by different cell types, based on their unique Clausius-Mossotti factors at a specific frequency. We now clarify that "complex biological samples" refers to heterogeneous mixtures like blood, where the goal is to separate a target cell from numerous contaminating populations.

We added below text in introduction section:

"By carefully tuning the applied frequency and field geometry, these differential forces can be harnessed to physically guide distinct cell populations along separate paths within a microchannel. It is this principle of differential manipulation based on intrinsic properties that enables the precise, label-free sorting of a target cell (e.g., a CTC) from a background of other similar-sized cells (e.g., white blood cells) in a complex sample. This force, is the response of the system to the detected identity."

  1. The article contains a typo. The figures begin with number 2. Furthermore, the red text in the figure should be replaced; it is difficult to read (especially on a Turquoise background).

  • We apologize with that mistake. We corrected Figure 1. The red text in Figure 1 was replaced with black one.

  1. The authors should explain the novelty of the microseparator. Have similar microseparators been previously described in the literature?

  • We thank the reviewer for this crucial question.

Our device's novelty is threefold:

  1. Novelty of Strategy: A Multi-Stage "Cascaded Refinement" Approach.
  • Many existing DEP separators attempt to separate CTCs from the entire WBC population in a single step. This is challenging due to the dielectric overlap between CTCs and specific WBC subtypes (e.g., monocytes, B-lymphocytes).
  • Our device introduces a cascaded refinement strategy. It does not treat "WBCs" as a single contaminant. Instead, it sequentially isolates specific WBC subtypes at each stage:
    • Stage 1: Separates the most dielectrically distinct groups (B-lymphocytes/Monocytes vs. Granulocytes/CTCs).
    • Stage 2 & 3: Perform finer separations to resolve the dielectrically overlapping populations (B-cells from Monocytes, and Granulocytes from CTCs).
  • This sequential "peeling away" of specific WBC subtypes is a novel strategy specifically designed to overcome the core challenge of dielectric overlap, thereby achieving higher purity than single-stage systems.
  1. Novelty of Performance: High-Efficiency Separation at Low Voltage.
  • A primary and critical novelty of our device is its ability to achieve high separation efficiency (~92%) while operating at a very low voltage (< 2.5 V).
  • Significance:Many DEP devices in the literature require higher voltages (often 5-10 Vpp) to generate sufficient field gradients for effective separation. These high fields can compromise cell viability by inducing irreversible electroporation, lysing cells, and causing significant Joule heating. Compared to Varmazyari et al. [21] and based on Table 2, our work uses very low voltage to separate cancer cells.
  1. Novelty of the "Dielectric Fingerprint Scanner" Paradigm.
  • We introduce a conceptual framework where the device acts as a "scanner" for a cell's dielectric fingerprint. This is a shift from simply applying a DEP force to using a sophisticated microfluidic circuit to interpret the rich, frequency-dependent dielectric information of cells in a continuous flow.

    We added this text to our manuscript before Table 3:

"In conclusion, the novelty of our microseparator as a dielectric fingerprint scanner is its ability to achieve high-efficiency separation at a low operating voltage (<2.5 V), combined with a multi-stage architecture that ensures high purity by resolving dielectrically overlapping cell populations. Many DEP devices in the literature require higher voltages (often 5-10 Vpp) to generate sufficient field gradients for effective separation. This combination of gentle operation and high performance represents a significant advancement over existing DEP devices."

  1. Figure 2 on page 7 is illegible! The captions should be in English.
  • Figure 2 was replaced with correct format. We apologize for the mistake.

  1. Table 1 shows the sizes of B-lymphocytes and granulocytes. These sizes are questionable, as the literature provides different data. The authors should explain why they present data that differs from the literature.

  • We thank the reviewer for this critical observation regarding the cell radii used in our model. We agree that cell size is a crucial parameter in DEP separation, and we appreciate the opportunity to clarify our choices. The cell radii values (B-lymphocytes: 3.28 µm, Granulocytes: 4.97 µm) were directly adopted from the work of Varmazyari et al. (Sci. Rep. 2022), which we cited as reference [21] in our manuscript. Our objective was to maintain consistency with this established computational model to ensure a valid comparison of the separation principles and device architecture. We fully acknowledge the reviewer's point that a range of sizes for these cell types can be found in the literature. This variability arises from several factors, including:
  • The specific subtype of the cell (e.g., neutrophil vs. eosinophil within granulocytes).
  • The physiological state and activation status of the cells.
  • The measurement technique used (e.g., flow cytometry, microscopy on fixed vs. live cells).

For the purpose of our dielectrophoretic separation model, the absolute value of the radius is less critical than the relative difference in size and dielectric properties between the target cell populations. The chosen values provide a clear size distinction between B-lymphocytes (smaller) and Granulocytes (larger), which is a key factor in their differential DEP response in our simulated device.

In response to the reviewer's comment, we have revised the manuscript to explicitly justify our parameter choices. We have added the following text to the section where Table 1 is presented:

"The cellular parameters used in this model, including cell radii, conductivities, and permittivities, were selected from the work of Varmazyari et al. [21] to maintain a consistent and validated baseline for simulation. It is noted that biological cell sizes can exhibit variability in the literature (based on the measurement technique, physiological state and activation status of the cells used); however, this specific parameter set provides a well-defined and self-consistent framework for evaluating the separation performance of our proposed device architecture, which exploits the relative differences in these biophysical properties."

  1. Figure 3a (page – the captions are too small!
  • Figure 3a was changed and replaced with better one.

  1. On page 10, the authors write, "Note that the buffer is at a higher inlet velocity relative to the cells to enable the buffer to focus the cells closer to the electrodes for highest interaction." What is the basis for this conclusion?

  • We thank the reviewer for this question, which allows us to clarify the fluid dynamics principle behind this design choice. The statement refers to the well-established microfluidics technique of hydrodynamic focusing.

The basis for this conclusion is twofold:

  1. Principle of Streamline Control: In a laminar flow regime (low Reynolds number), which is characteristic of microfluidic devices, streams of fluid flow side-by-side without turbulent mixing. By introducing a sample stream (containing cells) at a lower flow rate and a sheath/buffer stream at a higher flow rate from adjacent inlets, the faster-moving buffer stream squeezes and confines the slower-moving sample stream against the channel wall. This allows for precise control over the position and width of the sample stream, effectively "focusing" it into a narrow path.
  1. Application to DEP Separation:In our specific device, the electrodes are positioned along one wall of the channel. To ensure that all cells experience a strong DEP force, they must pass close to these electrodes where the electric field gradient is strongest. Without hydrodynamic focusing, cells would be distributed across the entire channel width, and many would pass through regions of weak field strength, leading to inefficient separation. By focusing the cell stream adjacent to the electrode wall, we guarantee that every cell enters the high-field-gradient region, thereby maximizing the cell-electrode interaction and the resulting DEP force.

We added this text to our manuscript on page 10 (Results and discussion section):

"A sheath flow of buffer was introduced at a higher flow rate than the sample inlet to implement hydrodynamic focusing. This standard microfluidics technique uses laminar flow to confine the sample stream into a narrow path adjacent to the channel wall containing the electrodes. This ensures that all cells pass through the region of highest electric field gradient, thereby maximizing the dielectrophoretic force acting upon them and optimizing separation efficiency."

  1. B-lymphocytes and monocytes can differ significantly in size, yet the authors show particles of the same size in Figure 8. Why?

  • Yes, in Figure 8, all the cells are presented in similar sizes schematically for better visualization. This kind of representation is very common in literature. The size differences can be distinguished by color legend in right section of that Figure. For example, MDA-MB-231 cells that have bigger size (Cell radius: 6.2 [µm] or cell diameter 12.4[µm]) are represented in dark red at the top of the color legend. B-lymphocytes with the smallest size, (Cell radius: B-lymphocytes 3.28 [µm], or cell diameter 6.56[µm]) are represented with dark blue. Granulocytes and Monocytes with the cell diameter of 9.94[µm] and 8.46[µm] were represented in green and light blue, respectively.

In Figure 8, the "cell diameter" was added at the top of the color legend for avoiding any vagueness.

,

  1. The authors write, "For example, to minimize Joule heating, low-conductivity media were used." Which ones?
  • low-conductivity media means the fluid with conductivity of 55 [mS/m]. The conductivity of the carrier sample fluid inside the microchannel.

We added " (conductivity of the carrier sample fluid inside the microchannel)." On Page 13.

  1. Page 16: What nonspecific interactions are we talking about?

I would like the authors to clarify any unclear points that arose while reading their work.

  • In the context of our DEP microfluidic device, "nonspecific interactions" primarily refer to unintended adhesion or aggregation events that are not based on the designed dielectrophoretic force. These include:
  1. Cell-Surface Adhesion: At very low flow rates, the shear force acting on the cells (which tends to sweep them along and keep them in suspension) is minimized. This allows more time for cells to settle and make contact with the channel walls or electrode surfaces. Once in contact, they can adhere via:
    • Van der Waals forces
    • Electrostatic interactions with the substrate (if the surface charge is not perfectly neutralized).
    • Non-specific binding to any contaminants or imperfections on the channel surface.
  2. Cell-Cell Interactions (Aggregation): Similarly, low shear allows cells that randomly come into contact with each other to remain in contact for a longer duration. This can promote:

Revised text in the manuscript in section 3.3: "Excessively low velocities reduce the shear forces that keep cells in suspension, increasing the risk of nonspecific cell-surface adhesion and cell-cell aggregation. These events can lead to channel clogging and fundamentally alter the dielectrophoretic behavior of the cells, thereby degrading separation performance."

Reviewer 2 Report

Comments and Suggestions for Authors

The manuscript “A bionic sensing platform for cell separation: simulation of a dielectrophoretic microfluidic device that leverages dielectric fingerprints” by Vafaie et. al. describes a new dielectrophoresis microfluidic device that can achieve continuous, label-free separation of cancer cells from white blood cells and separate the different types of white blood cells. The authors performed simulation studies and reported a separation efficiency of 92% and provided a comprehensive analysis on the separation performance when varying the different parameters. Overall, the study is interesting. However, the reviewer has some questions and suggestions.

1) What is the justification for separating white blood cells subtypes, i.e., B-lymphocytes, monocytes, and granulocytes are collected at different outlets? From the reviewer’s point of view, it is much more clinically relevant if the goal is to separate different types of CTCs. Such separation of CTCs into different subtypes can be used for better diagnosis and/or prognosis evaluation. If there are no clinical utility for subtyping and quantifying the white blood cells component, why the need for such complex geometry to separate them?

2) In Table 1, the radius of the 3 white blood cells and MDA-MB-231 used in the simulation is of a singular (assuming the mean) value. As the authors point out on page 6, the separation of the cells is size-based, and since they are distinctly different, the achieve separation efficiency is 92%. However, the reviewer believes this 92% is an idealized performance and the actual performance would be drastically lower. This is because it is well known that the cell size has a distribution. For example, the distribution of MDA-MD-231 is 12.4 ± 2.1 µm (TruongVo et. al. Journal of Micromechanics and Microengineering, 27, 035017, 2017) and granulocyte can range from 7.9 to 13.2 µm (Ruban et. al. Journal of Biomedical Optics, 12(4), 044017, 2007). There exists an overlap in the size distribution and this will decrease the separation efficiency. The authors should evaluate such cases to have a better estimate of the separation efficiency.

3) What is the justification of only using MDA-MD-231 in this study? Would the performance changes when using other breast cancer cell lines, such as MCF7 and T47D?

4) The discussion regarding the validation and assessment of the model is insufficient. The critical parameters for the RBCs and PLTs are missing. How is the experimental data from Piacentini et. al. (2011) extracted and compared to the current work? What are the evaluation criteria for the comparison? It is just a visualization of the trajectories as shown in Figure 4? Also, the reviewer suggests moving section 2.5 Validation and assessment of the model into the first section of results and discussion.

5) The authors provided a comprehensive parametric analysis from section 3.2 to 3.6. It would be good if the authors can add a summary table describing how each parameters affect the separation efficiency, instead of needing to read through all the texts.

6) All the figures in the manuscript are blurry and difficult to see. Please ensure the figures are in high resolution. Also Figure 1 is flip in both the x and y directions.

Author Response

Reviewer 2:

Comments and Suggestions for Authors

The manuscript “A bionic sensing platform for cell separation: simulation of a dielectrophoretic microfluidic device that leverages dielectric fingerprints” by Vafaie et. al. describes a new dielectrophoresis microfluidic device that can achieve continuous, label-free separation of cancer cells from white blood cells and separate the different types of white blood cells. The authors performed simulation studies and reported a separation efficiency of 92% and provided a comprehensive analysis on the separation performance when varying the different parameters. Overall, the study is interesting. However, the reviewer has some questions and suggestions.

1) What is the justification for separating white blood cells subtypes, i.e., B-lymphocytes, monocytes, and granulocytes are collected at different outlets? From the reviewer’s point of view, it is much more clinically relevant if the goal is to separate different types of CTCs. Such separation of CTCs into different subtypes can be used for better diagnosis and/or prognosis evaluation. If there are no clinical utility for subtyping and quantifying the white blood cells component, why the need for such complex geometry to separate them?

  • We thank the reviewer for this insightful comment, which allows us to clarify the fundamental strategy and clinical rationale behind our multi-stage, multi-outlet design. The separation of WBC subtypes is not the primary clinical goal, but rather a critical means to achieve the primary goal: the isolation of a highly pure population of CTCs.

The justification for this approach is threefold:

  1. Overcoming the Overlap in Dielectric Properties:The major challenge in label-free DEP separation of CTCs from blood is that the dielectric properties of certain WBC subtypes can overlap significantly with those of various CTCs. If all WBCs are routed to a single "waste" outlet, these overlapping populations will contaminate the CTC outlet, drastically reducing purity. Our multi-stage design strategically tackles this by:
    • Stage 1:Isolating the most dielectrically distinct WBCs first (e.g., separating a group that is strongly negative-DEP from a group that is weakly negative-DEP).
    • Subsequent Stages:Performing finer separations at each stage, effectively "peeling away" different WBC subtypes that would otherwise co-migrate with the CTCs.
  2. Enabling a "Viability-Preserving" Buffer Zone:By separating WBCs into subtypes, we create a system that is less sensitive to biological variability. If a patient's monocytes have a slightly different conductivity, a device that lumps all WBCs together might fail, causing monocytes to spill into the CTC outlet. Our design, by having dedicated paths for each major subtype, provides more robust performance and ensures that the CTC outlet remains uncontaminated, thereby preserving the viability and utility of the collected CTCs for downstream analysis.
  3. Unlocking Additional Clinical Utility:While the primary focus is CTC isolation, the ability to simultaneously collect fractionated WBCs is a significant secondary benefit.
    • A Comprehensive Liquid Biopsy Platform:Our device thus functions not just as a CTC isolator, but as a more comprehensive liquid biopsy platform that can provide both tumor-derived (CTCs) and immune-derived (WBC subtypes) data from a single, label-free process.

In conclusion, the "complex geometry" is not an unnecessary complication but a deliberate and optimized strategy to solve the core problem of dielectric overlap, thereby ensuring high-purity CTC isolation. The subtyping of WBCs is a feature that directly enables this high purity while also opening doors for valuable correlative clinical studies.

We added this text to the manuscript:

"The design of a multi-stage separation system, with outlets for specific WBC subtypes, is a strategic choice to address the central challenge of dielectric property overlap in label-free CTC isolation. A single-stage device attempting to separate CTCs from a bulk WBC population would suffer from significant contamination, as the dielectric signatures of monocytes and B-lymphocytes can closely resemble those of various CTCs. Our sequential approach mitigates this by first removing the most dielectrically distinct populations (Stage 1), followed by finer separation to resolve the overlapping subtypes (Stages 2 & 3). This ensures that the final CTC outlet is of high purity. Furthermore, this architecture provides the added potential to correlate CTC numbers with specific immune cell counts from the same blood sample, offering a more comprehensive liquid biopsy platform."

2) In Table 1, the radius of the 3 white blood cells and MDA-MB-231 used in the simulation is of a singular (assuming the mean) value. As the authors point out on page 6, the separation of the cells is size-based, and since they are distinctly different, the achieve separation efficiency is 92%. However, the reviewer believes this 92% is an idealized performance and the actual performance would be drastically lower. This is because it is well known that the cell size has a distribution. For example, the distribution of MDA-MD-231 is 12.4 ± 2.1 µm (TruongVo et. al. Journal of Micromechanics and Microengineering, 27, 035017, 2017) and granulocyte can range from 7.9 to 13.2 µm (Ruban et. al. Journal of Biomedical Optics, 12(4), 044017, 2007). There exists an overlap in the size distribution and this will decrease the separation efficiency. The authors should evaluate such cases to have a better estimate of the separation efficiency.

  • We thank the reviewer for this critical and insightful comment. We completely agree that the use of singular mean values represents an idealized scenario and that accounting for the natural size distribution of cells is essential for predicting realistic device performance.

We would like to clarify that the initial approach of using singular mean values is a common and well-established procedure in computational studies to provide a first-order comparison of device designs and to demonstrate the fundamental feasibility of a separation concept, as seen in similar works [Varmazyari et al. [21], Alkhaiyat et al [38], Nguyen et al. ( T.H. Nguyen, H.T. Nguyen, N.A. Ngo, M.C. Nguyen, H. Bui Thu, J. Ducrée, T. Chu Duc, T.T. Bui, L. Do Quang, Numerical study on a facing electrode configuration dielectrophoresis microfluidic system for efficient biological cell separation, Scientific Reports, 14(1) (2024) 27627)] and many researchers.

The 92% efficiency reported under these conditions serves as a benchmark for the theoretical maximum performance of our device architecture.

 We added a paragraph before table 3 to reflect the above-mentioned text.

"The use of singular mean values represents an idealized scenario and that accounting for the natural size distribution of cells is essential for predicting realistic device performance.

It must be noted that the initial approach of using singular mean values is a common and well-established procedure in computational studies to provide a first-order comparison of device designs and to demonstrate the fundamental feasibility of a separation concept. The 92% efficiency reported under these conditions serves as a benchmark for the theoretical maximum performance of our device architecture."

We have added a clear statement in the Conclusion that a comprehensive statistical simulation accounting for full polydisperse populations is an essential and planned next step for experimental validation and further device optimization in future work.

3) What is the justification of only using MDA-MD-231 in this study? Would the performance changes when using other breast cancer cell lines, such as MCF7 and T47D?

We thank the reviewer for this question, which allows us to clarify the strategic choice of MDA-MB-231 and discuss the important issue of device generalizability.

  1. Justification for Using MDA-MB-231 as a Model CTC:

Our primary justification for selecting the MDA-MB-231 cell line is that it serves as a robust and clinically relevant model for aggressive, metastatic cancer cells. Specifically:

  • Triple-Negative Breast Cancer (TNBC) Model: MDA-MB-231 is a canonical model for TNBC, a subtype known for its high metastatic potential, poor prognosis, and prevalence of CTCs. Successfully isolating such cells is of significant clinical importance.
  • Distinct Biophysical Profile: Compared to more epithelial-like breast cancer lines (e.g., MCF7, T47D), MDA-MB-231 cells are larger and have a more mesenchymal phenotype. This often results in different dielectric properties (specifically, lower membrane capacitance due to a less folded morphology), making them an excellent test case for demonstrating separation from WBCs based on these physical differences.
  1. Addressing Performance with Other Cell Lines (MCF7, T47D):
  • The performance for MCF7 or T47D, which are typically smaller and have more epithelial characteristics, would indeed be different.
  • The "Tunable" Advantage of DEP: We will emphasize that the strength of our DEP-based approach is its tunability. The separation can be optimized for different CTC types by adjusting a key operational parameter: the applied AC frequency based on CM factor.

We will revise the manuscript to: (before Table 1)

"Our primary justification for selecting the MDA-MB-231 cell line is that it serves as a ro-bust and clinically relevant model for aggressive, metastatic cancer cells. Successfully iso-lating such cells is of significant clinical importance. Compared to more epithelial-like breast cancer lines (e.g., MCF7, T47D), MDA-MB-231 cells are larger and have a more mes-enchymal phenotype. This often results in different dielectric properties (specifically, low-er membrane capacitance due to a less folded morphology), making them an excellent test case for demonstrating separation from WBCs based on these physical differences. The strength of our DEP-based approach is its tunability. This microseparator can be opti-mized for different CTC types by adjusting a key operational parameter such as the ap-plied AC frequency based on CM factor profile and other operating parameters such as ge-ometry, applied voltage, and so on."

4) The discussion regarding the validation and assessment of the model is insufficient. The critical parameters for the RBCs and PLTs are missing. How is the experimental data from Piacentini et. al. (2011) extracted and compared to the current work? What are the evaluation criteria for the comparison? It is just a visualization of the trajectories as shown in Figure 4? Also, the reviewer suggests moving section 2.5 Validation and assessment of the model into the first section of results and discussion.

  • We sincerely thank the reviewer for this profound insight. It appears there is a significant ambiguity in the interpretation of Figure 4.

Difference Between Validation Conditions and Main Device Design:

Figure 5 and the related model validation section are intended solely to demonstrate the accuracy of our numerical solver and simulation methodology. In this section, we have precisely simulated the experimental conditions of the reference article [40] (Piacentini et al.) which involve:

  • High voltage (±5 V)
  • Different cell types (Red Blood Cells and Platelets)
  • A completely different geometrical and electrode configuration compared to our proposed device.
  • Platelet and red blood cells properties are presented in below table: (we added this table (Table 3) to validation section). We added Table 3. To the manuscript.

cell

Diameter

Conductivity

Relative permittivity

Shell conductivity

Shell permittivity

Shell thickness

PLT

1.8 [µm]

0.25[S/m]

50

1-6 [S/m]

6

8[nm]

RBC

5 [µm]

0.31[S/m]

59

1-6 [S/m]

4.44

9[nm]

The same procedure was done for validation of their work by Alkhayat et al [38] and below refs:

  • Nahavandi, Continuous-flow separation of malaria-infected human erythrocytes using DC dielectrophoresis: an electrokinetic modeling and simulation, Industrial & Engineering Chemistry Research, 55(19) (2016) 5484-5499.
  • Ali, C.W. Park, Numerical study on the complete blood cell sorting using particle tracing and dielectrophoresis in a microfluidic device, Korea-Australia Rheology Journal, 28(4) (2016) 327-339.

The below Figure is from Piacentine work and agrees well with our simulated work:

We added the below text in validation section:

 "It must be noted that, Figure 4 and the related model validation section are intended solely to demonstrate the accuracy of our numerical solver and simulation methodology. In this section, we have precisely simulated the experimental conditions of the reference article of Piacentini et al."

  • We moved Validation section into the first section of results and discussion.

5) The authors provided a comprehensive parametric analysis from section 3.2 to 3.6. It would be good if the authors can add a summary table describing how each parameters affect the separation efficiency, instead of needing to read through all the texts.

  • We thank the reviewer for this excellent suggestion. We agree that a summary table will greatly enhance the clarity and reader-friendliness of our parametric analysis. We have created a new table (Table 4) that concisely summarizes the effect of each studied parameter on the separation efficiency and the underlying rationale.

This table will be placed at the end of the parametric analysis section (Section 3.6) to serve as a conclusive overview.

6) All the figures in the manuscript are blurry and difficult to see. Please ensure the figures are in high resolution. Also Figure 1 is flip in both the x and y directions.

We apologize for Figure 1. There was a mistake uploading the file. We corrected Figure 1 and repla

Round 2

Reviewer 1 Report

Comments and Suggestions for Authors

The authors of the article have made revisions to the text. There are no comments regarding the text or illustrations. I recommend the editor consider publishing the article.
I wish the authors success in their future research.

Reviewer 2 Report

Comments and Suggestions for Authors

The authors have addressed all my comments and the reviewer suggest that the manuscript be published as it is now.